# College Students’ eMental Health Literacy and Risk of Diagnosis with Mental Health Disorders

**DOI:** 10.3390/healthcare10122406

**Published:** 2022-11-30

**Authors:** Eileen Cormier, Hyejin Park, Glenna Schluck

**Affiliations:** College of Nursing, Florida State University, Tallahassee, FL 32306, USA

**Keywords:** eMental health literacy, mental health disorders, college students, digital mental health literacy

## Abstract

Background: This study investigated college students’ eMental health literacy (eMHL), knowledge of common mental disorders and risk of being diagnosed with a mental health disorder and compared their knowledge of mental disorders and concurrent risk of diagnosis with high and low eMHL; Method: A total of 123 college students completed an online survey through Amazon’s Mechanical Turk (MTurk). Data were analyzed using descriptive statistics and chi-square tests; Results: eMental health literacy scores were higher when students had a history of prior mental health problems, were female, and graduate students. College students with high eMHL were more likely to recognize symptoms, recommend professional help, and be at lower risk for common mental health disorders compared to low eMHL students; Conclusions: eMHL was associated with mental health status and demographic variables. Level of eMHL was associated with knowledge of mental disorders and risk of diagnosis. Implications: The results highlight the need for targeted interventions to enhance eMHL of college students, support mental health resilience and prevent mental health disorders.

## 1. Introduction

Mental health problems among college students have become a growing concern in the US [1]. More than 25% of college students have been diagnosed or treated by a professional for a mental illness; 64% of college students no longer attend college due to mental health problems [2]. Recent college student data suggests rising rates of clinically significant mental health problems such as anxiety, depression, suicide, and substance use disorders [3,4]. However, despite the availability of counseling services that includes telehealth and the widespread promotion of such services, many college students fail to seek help due to fear of personal stigma, not perceiving treatment as urgent or essential, and being unaware services are available [5,6]. Delayed help-seeking or underutilization of available sources of care have a direct negative impact on college students’ mental health, quality of life, and academic achievement [7].

Mental health literacy (MHL) plays an important role in both the recognition of mental health problems and help-seeking for those problems [8]. MHL refers to the knowledge and beliefs about mental health disorders which aid in their recognition, management, or prevention [9]. Research findings provide support for a relationship between MHL and mental health services use [10,11,12,13,14]. College students with low MHL are less likely to seek help and receive treatment than students with high MHL [12]. Kim, Yu, and Kim [8] reported that MHL had a direct effect on attitudes toward help seeking and help-seeking intentions through stigma.

With advanced digital technology, a tremendous amount and variety of mental health-related information can be accessed with ease, anonymously, and at relatively low cost. Many colleges students seek mental health information for decision-making guidance, to explore service options, or seek further professional help on the internet [15]. They require abilities to identify and retrieve relevant online information sources, evaluate the quality of the online information and its applicability to their query, and mindfully use the information to make sound mental health decisions [16]. Norman and others hold that the influencing factors of online Information searching are positively correlated with electronic health (eHealth) literacy at the level of individual ability [17].

Electronic mental health literacy (eMHL) is an emerging concept of scholarly interest that is seen as an extension of MHL in the digital era [18]. eMHL is conceptually aligned with mental health literacy (MHL). eMHL is defined as “the degree to which individuals obtain, process, and understand online mental health information and services needed to aid their recognition, management, or prevention of mental health issues” [19] (p. 305). College students as a population need mental health information literacy skills at a critical stage in their psychosocial development to make mental health decisions that have lifelong impact [20]. However, previous studies indicate that college students have inadequate online health information literacy skills [21,22] and lack sufficient mental health literacy [23,24].

Although eMHL is a significant influencing factor in college students’ mental health [19,25], studies on the association of eMHL with knowledge of mental health disorders and concurrent risk for diagnoses among college students is remarkably limited. Thus, the current study hypothesized that there are differences in knowledge of mental health disorders and risk of a mental health disorder diagnosis between college students with low and high eMHL. The specific aims of this study were to (1) assess college students’ eMHL, their knowledge of common mental health disorders, and their risk of being diagnosed with a mental health disorder, and (2) compare knowledge of common mental health disorders and risk of a mental health disorder diagnosis in college students with high and low eMHL.

### 1.1. Literature Review

Due to the global high prevalence of mental disorders, there is an increase in studies of influencing factors on mental health such as MHL, eMHL, and help-seeking for mental health problems. MHL in the research literature is often conceptualized as the awareness of problems as evidenced by correct identification of mental health problems, and attitudes towards seeking appropriate help [8,9]. Based on this definition, many studies have focused on the assessment of MHL level and its association with attitudes towards help seeking in different populations and countries [26,27,28]. A longitudinal study of nursing students’ MHL in Australia reported significant improvements in MHL regarding schizophrenia as students progressed in their program; they became more knowledgeable about the potential benefits of assistance from health professionals, mental health-specific medications, and non-pharmacological therapies [27]. Another study by Reavley et al. [28] examined depression MHL of university students and staff and reported being female, more highly educated, and older in age were associated with accurate recognition of the disorder. General practitioners were designated as the most likely source of help followed by seeking help from friends, parents, and helplines [28]. In another study of university students from different ethnic backgrounds, Chinese female students had a comparatively better knowledge of the symptoms of depression compared to their Malay and Indian counterparts [29]. In a study of adolescents, grade level and socioeconomic factors were not associated with MHL but being female and having prior experience with mental health problems and help-seeking from difference sources [30] were associated with higher MHL.

MHL has been shown to be an influencing factor in mental health outcomes [31,32,33]. In a study of young adults, low MHL level was associated with compromised mental health status, particularly higher levels of depression [31]. MHL has also been linked to self-care skills for alleviating mental health concerns and improving public mental health in general [32]. The findings of one study indicated higher MHL had a positive and significant correlation with a range of health-promoting behaviors [33].

### 1.2. Electronic Mental Health Literacy (eMHL) and Mental Health Outcome

eMHL is conceptually aligned with mental health literacy (MHL) and includes digital literacy and mental health literacy. Despite the successfully engagement of digital technology in mental healthcare, there are considerable barriers to and inequities in digital MH care due to low digital literacy [34]. Individuals experiencing mental illness may own smartphones but may not have the technical skills to participate in the digital world and to utilize health-related technology to its fullest potential [35]. Low digital skills in mental health influence in mental health outcomes. For example, some studies showed that lack of digital literacy skills have negative consequences for people with mental health disorders, because they fail to identify useful and safe web-based resources or tools with the potential to support their self-management [36,37].

Digital mental health literacy skills, eMental health literacy skills, are of clear importance for people living with mental health disorders. There are limited studies on eMental health literacy and mental health outcomes. Further studies are necessary to improve eMHL among college students to achieve and maintain positive mental health.

## 2. Materials and Methods

### 2.1. Data Collection

*Study design and recruitment*: A cross sectional design was used for this study. College students in the U.S. were invited to participate via Amazon’s Mechanical Turk (MTurk) since mental health problems in this population have become a growing concern in the U.S. MTurk is an online, web-based tool that allow researchers to recruit large, representative samples of research participants [38]. MTurk is an efficient, reliable, and cost-effective tool for generating sample responses that are largely comparable to those collected via more conventional means [39].

*Study sample*: Participants in this study were restricted to college students aged 18 and older who were able to speak and read in English in the U.S. A total of 123 participants completed the surveys using MTurk after completing the online informed consent. Participants who completed the surveys received $2.00 in compensation. The survey offered on MTurk was linked to, and automatically saved in, Qualtrics. Participants were asked to complete questionnaires related to (1) demographic characteristics; (2) internet use and eMHL; (3) knowledge of common mental health disorders among college students; and (4) students’ risk for common mental health disorders.

### 2.2. Instruments

#### 2.2.1. eHEALS: eMental Health Literacy

For the purposes of this study, a modified eHealth literacy scale (eHEALS) [40] was used to assess college students’ eMHL. The eHEALS is an 8-item self-report tool with a five point-Likert scale ranging from 1 (strongly disagree) to 5 (strongly agree). The 8 items measure knowledge of what online health information is available, where one can find helpful health resources, how to access this information, and skills to evaluate or discern the quality of online health information. Two additional items that were not computed in the total score of eHEALs were included to assess perceived usefulness of the internet for making health decisions and the importance of being able to access online health resources [40].

#### 2.2.2. Mental Health Literacy

Six short vignettes depicting college students experiencing symptoms of major depressive disorder (a male and female scenario), anxiety disorder (a female with social anxiety disorder and a male with obsessive compulsive disorder) and schizophrenia spectrum disorders (a male and female scenario) were used to assess college students’ knowledge of common mental disorders. Each participant was presented with three vignettes and asked to respond to a set of six multiple-choice questions based on Jorm’s model [41] of MHL to identify their knowledge and beliefs about causes of the disorder, self-help interventions, sources of professional help, and how to seek mental health information.

#### 2.2.3. DSM-5 Self-Rated Level 1 Cross-Cutting Symptom Measure—Adult

The DSM-5 Level 1 Cross-Cutting Symptom Measure was used to measure college students’ risk for common mental health disorders. The DSM-5 Level 1 consists of 23 items which capture 13 psychiatric domains: depression, anger, mania, anxiety, somatic symptoms, suicidal ideation, psychosis, sleep problems, memory, repetitive thoughts and behaviors, dissociation, personality functioning, and substance use. Each item is rated on a five-point scale (0 = none or not at all; 1 = slight or rare, less than a day or two; 2 = mild or several days; 3 = moderate or more than half the days; and 4 = severe or nearly every day). The participants were asked how much they had been bothered by each symptom during the past 2 weeks. A score of 2 or higher for any item in a domain except substance use (score 1 or higher) represents a risk of mental health disorders [42].

### 2.3. Data Analysis

Descriptive statistics were computed to summarize demographic data, eMHL of college students, college students’ knowledge of mental health disorders, and risk for common mental health disorders. Chi-square tests of association were used to test the associations between college students’ eMHL and their knowledge of common mental disorders and their risk of being diagnosed with a mental health disorder.

## 3. Results

### 3.1. Demographics

Demographic characteristics of the 123-college student participants are presented in Table 1. The average age was 32 (SD = 11) with approximately half of the students identifying as female (n = 63, 51.2%). The majority were Caucasian (n = 86, 69.9%), attended school full time (n = 88, 71.5%), had never been married (n = 64, 52%), and spoke English at home (n = 116, 94.3%). The students either lived with a partner and/or children (n = 49, 39.8%), off campus (n = 46, 37.4%), with parents (n = 18, 14.6%), or in university residential housing (n = 10, 8.1%). They reported occasional (n = 58, 47.8%), frequent (n = 28; 22.8%) or constant (n = 20; 16.3%) financial stress; the majority worked more than 20 h per week (n = 83, 67.5%). Most indicated they had health insurance (n = 100, 81.3%) and rated their health as good or better (n = 109, 88.6%). Approximately half (n = 63, 51.2%) had received help for a mental health problem in the past, but most had never tried to get help for a mental health concern for a family member or friend (n = 76; 61.8%). College students who had received help for a mental health problem (n = 28, 44.4%) were more likely to report accessing help for a mental health concern for a family member or friend compared to students who had not (n = 19, 31.7%).

### 3.2. eMental Health Literacy Skills

The highest proportion of student participants reported that they used the internet 4 to 10 h per day (n = 55, 44.7%) and from home (n = 107, 87%). The majority also accessed the internet from their smart phones (n = 116, 94.3%) and had been using the internet for over five years (n = 113, 91.9%). College students were designated as having high eMHL Skills if their eHEALS scores were above the mean score of 31.3 (SD 6.0). Approximately half of the college students displayed low eMHL skills (n = 62, 50.4%), yet 78% (n = 96) indicated that they agreed or strongly agreed with the statement. “I know how to use mental health information I find on the Internet to help decision making” (Table 1).

### 3.3. Knowledge of Mental Health Disorders

College students were asked a series of questions corresponding to three out of six mental health scenarios: Scenario 1 included a male by the name of Shawn (1A) and a female called Mina (1B) with major depressive disorder; Scenario 2, both anxiety disorders, included a female named Rachel with an social anxiety disorder (2A) and a male called Justin with obsessive compulsive disorder (2B); and Scenario 3 included a female, Caroline (3A), and a male, Max (3B), exhibiting symptoms of schizophrenia spectrum disorder (Table 2). The students could select either A or B for each of the three scenarios. For Scenario 1 (depression), 84 (68.3%) students selected the male scenario (Scenario 1A) and 39 (31.7%) selected the female scenario (Scenario 1B). For Scenario 2 (anxiety), 62 (50.4%) students selected the female scenario (Scenario 2A) and 61 (49.6%) selected the male scenario (Scenario 2B). For Scenario 3 (schizophrenia), 76 (61.8%) selected the female scenario (Scenario 3A) and 47 (38.2%) selected the male scenario (Scenario 3B). A total of 106 college students (86.2%) correctly diagnosed at least two out of three scenarios they were given. Approximately 50% of these (n = 54, 43.9%) also correctly diagnosed the third scenario (Table 2).

#### 3.3.1. Scenario: Depression

When considering Scenario 1A and 1B together, 56.9% (n = 70) correctly identified major depressive disorder as the diagnosis. The primary cause of the problem, according to the students surveyed, was environmental (n = 63, 51.2%; Scenario 1A: n = 34, 40.5%; Scenario 1B: n = 29, 74.4%,) versus personal vulnerability (n = 33, 26.8%; Scenario 1A: n = 26, 31.0%; Scenario 1B: n = 7, 17.9%) or mental illness (n = 21, 17.1%; Scenario 1A: n = 18, 21.4%; Scenario 1B: n = 3, 7.7%). When asked what actions were needed to deal with the problem, 40.7% (n = 50) of college students indicated they should talk to a counselor on campus (n = 50; Scenario 1A: n = 31, 36.9%; Scenario 1B: n = 19, 48.7%;), 28.5% recommended a psychiatrist or psychologist (n = 35; Scenario 1A: n = 26, 31.0%; Scenario 1B: n = 9, 23.1%) and 27.6% indicated they should talk with family and friends (n = 34; Scenario 1A: n = 25, 29.8%; Scenario 1B: n = 9, 23.1%). Sixty percent of college students advocated professional help (n = 74; Scenario 1A: n = 54, 64.3%; Scenario 1B: n = 20, 51.3%;), favoring a psychologist (n = 36, 48.6%; Scenario 1A: n = 23, 27.4%; Scenario 1B: n = 13, 65.0%) or a psychiatrist (n = 22, 29.7%; Scenario 1A: n = 23, 27.4%; Scenario 1B: n = 3, 15.0%). When asked where the student could go to find more information about the problem, 49.6% of respondents selected mental health websites (n = 61; Scenario 1A: n = 39, 46.4%; Scenario 1B: n = 22, 56.4%) and 41.5% selected someone who has experience with a similar problem (n = 51; Scenario 1A: n = 37, 44.0%; Scenario 1B: n = 14, 35.9%).

#### 3.3.2. Scenario: Social Anxiety Disorder and Obsessive Compulsive Disorder

Social anxiety disorder (SAD) and obsessive compulsive disorder (OCD) were correctly identified by 84.6% of college students. Mental illness was selected as the primary cause of the problem by 54.5% (n = 67; Scenario 2A: n = 25, 40.3%; Male Scenario 2B: n = 42, 68.9%); 26% indicated personal vulnerability was responsible (n = 32; Female Scenario A: n = 24, 38.7%; Male Scenario 2B: n = 8, 13.1%). When asked what actions should be taken to deal with the problem, most recommended the student should talk to a psychiatrist or psychologist (n = 86, 69.9%; Scenario 2A: n = 39, 62.9%; Scenario 2B: n = 47, 77.0%). The vast majority indicated the student should seek professional help (n = 102, 83.9%; Scenario 2A: n = 49, 79.0%; Scenario 2B: n = 53, 86.9%) and that the help should come from either a psychiatrist (n = 47, 46.1%; Scenario A: n = 19, 38.8%; Scenario 2B: n = 28, 52.8%) or a psychologist (n = 46, 45.1%; Scenario A: n = 24, 49.0%; Scenario 2B: n = 22, 41.5%). Over half of respondents advocated seeking out mental health websites (n = 68, 55.3%; Scenario 2A: n = 35, 56.5%; Scenario B: n = 33, 54.1%) or talking with someone who had experience with a similar problem to find more information about the problem (n = 42, 34.1%; Scenario 2A: n = 21, 33.9%; Scenario 2B: n = 21, 34.4%).

#### 3.3.3. Scenario: Schizophrenia

Schizophrenia was correctly identified by most college student participants. The majority (78%; n = 96) indicated the primary cause of the problem was mental illness (Scenario A 3: n = 56, 73.7%; Scenario 3B: n = 40, 85.1%). Most advised seeing professional help (n = 112, 91.1%; Scenario 3A: n = 70, 92.1%; Scenario 3B: n = 42, 89.4%) from a psychiatrist (n = 78, 63.4%; Scenario 3A: n = 47, 67.1%; Male Scenario 3B: n = 31, 73.8%) or psychologist (n = 30, 24.4%; Scenario 3A: n = 21, 30.0%; Scenario 3B: n = 9, 21.4%). When asked where the student could go to find more information about the problem, respondents recommended mental health websites (n = 59, 48.0%; Scenario 3A: n = 34, 44.7%; Scenario 3B: n = 25, 53.2%) or talking to someone who had experience with a similar problem (n = 57, 46.3%; Scenario 3A: n = 40, 52.6%; Scenario 3B: n = 17, 36.2%).

### 3.4. Risk of Mental Health Disorder Diagnosis

College students were most at risk for anxiety (n = 70, 56.9%), substance abuse (n = 60, 48.8%), depression (n = 52, 42.3%), sleep problems (n = 49, 39.8%), and mania (n = 49, 39.8%). They were least at risk for dissociation (n = 17, 13.8%), psychosis (n = 26, 21.1%), and memory problems (n = 27, 22.0%) (Table 3).

### 3.5. Knowledge of Mental Disorders by Levels of eMHL

College students with high MHL correctly identified the mental health disorders more often (Depression: n = 40, 65.6%; SAD/OCD: n = 57, 93.4%; Schizophrenia: n = 54, 88.5%) than college students with low MHL (Depression: n = 30, 48.4%; Anxiety/OCD: n = 47, 75.8%; Schizophrenia: n = 49, 79.0%; Table 4). The primary cause of depression was most frequently identified as either environmental or personal vulnerability by both high (environmental: n = 31, 50.8%; personal vulnerability: n = 13, 21.3%) and low (environmental: n = 32, 51.6%; personal vulnerability: n = 20, 32.3%) literacy college students although college students with high literacy were less likely to indicate personal vulnerability as a cause. When asked what actions could be taken to deal with the problem, more college students with high MHL (n = 28, 45.9%) recommended talk to a counselor on campus compared to college students with low MHL (n = 22, 35.5%). More students with high MHL also indicated professional help was needed (high: n = 42, 68.9%; low: n = 32, 51.6%). When asked to specify the type of professional help needed, college students with high MHL selected a psychologist more often (n = 23, 54.8%) than college students with low MHL (n = 13, 40.6%). College students with low MHL were more likely to indicate the professional help should come from a primary care provider (low: n = 7, 21.9%; high: n = 5, 11.9%). College students with high MHL advocated additional help on mental health websites (n = 39, 63.9%) whereas college students with low MHL endorsed talking to someone with similar experience (n = 32, 51.6%).

The primary cause of anxiety/OCD was most frequently identified by college students with high MHL as mental illness (n = 38, 62.3%) whereas college students with low MHL listed either mental illness (n = 29, 46.8%) or personal vulnerability (n = 21, 33.9%) as the cause. When asked what actions could be taken, college students with high and low MHL agree that the person should talk to a psychologist or psychiatrist (low: n = 39, 62.9%; high: n = 47, 77.0%). A higher proportion of college students with high MHL indicated professional help was needed compared to those with low literacy (low: n = 47, 75.8%; high: n = 55, 90.2%). Additionally, although all of the college students advised professional help for anxiety or OCD from a psychologist or psychiatrist, more college students with high MHL favored a psychiatrist (high: n = 27, 49.1%; low: n = 20, 42.6%) versus a psychologist for low MHL students (low: n = 23, 48.9%; high: n = 23, 41.8%). Both the high and low literacy groups indicated that additional information could be obtained from mental health websites (low: n = 31, 50.0%; high: n = 37, 60.6%) or someone with similar experience (low: n = 23, 37.1%; high: n = 19, 31.1%).

The primary cause of schizophrenia was mental illness for both high and low MHL college students (low: n = 46, 74.2%; high: n = 50, 82.0%). When asked what actions could be taken, college students in both groups agreed that professional help should be sought (low: n = 54, 87.1%; high: n = 58, 95.1%) from a psychiatrist or psychologist (low: n = 55, 88.7%; high: n = 55, 90.2%). More college students with high MHL preferred a psychiatrist (high: n = 43, 74.1%; low: n = 35, 64.8%) while a higher proportion of college students with low MHL selected a psychologist (low: n = 17, 31.4%; high: n = 13, 22.4%). College students with high MHL were more likely to advise using mental health websites (high: n = 34, 55.7%; low: n = 25, 40.3%) to obtain additional information compared to those with low MHL who favored seeking out someone with similar experience (high: n = 25, 41.0%; low: n = 32, 51.6%) (Table 4).

### 3.6. Risk of Diagnosis by Levels of eMHL

*Chi*-square tests of association were computed to determine if there was an association between literacy (high/low) and risk of diagnosis (yes/no) for each of the 13 domains on the DSM-5 Self-Rated level 1 Cross-Cutting Symptom measure. In order to maintain a family-wise error rate of 0.05, a Bonferroni correction was applied. With 13 tests performed, tests were considered significant if the *p*-value was smaller than 0.004 (0.05/13). College students with low eMHL were more likely than those with high eMHL to be at risk for suicidal ideation (low: n = 21, 33.9%; high: n = 11, 18.0%). Overall, the percentage of college students at risk for diagnosis across all 13 domains was 3–16% higher for those with low eMHL compared to those with high eMHL (Table 5).

## 4. Discussion

The current study examined level of eMHL, knowledge of mental health disorders and risk of mental health disorders, the relationship between eMHL and knowledge of mental health disorders and risk of mental health disorders. Very little is known about college students’ eMHL and how it is related to knowledge level and risk of mental health disorders.

### 4.1. eMental Health Literacy

eMHL scores in this study were higher when students had a history of prior mental health problems, was female as opposed to male, and was a graduate versus undergraduate student. Similarly, Gorczynski et al. [10] found women, and those with a history of mental disorders indicated high levels of mental health literacy. In Reavley et al.’s study [28], being female with a higher education level was associated with higher mental health literacy.

Another study also reported undergraduate men had lower MHL and low intentions to seek professional care compared to graduate students [12]. Others found that MHL among international students is affected by foreign language skills [43,44]. Studies of eHealth literacy among college student concur that social demographic characteristics such as age, gender, and education influence levels of eHealth literacy [45,46].

### 4.2. Knowledge of Mental Health Disorders

Knowledge of common mental health disorders (depressive disorder, social anxiety disorder/obsessive compulsive disorder, schizophrenia spectrum disorder) among college students was assessed based on responses to questions related to three of six possible vignettes. Most college students correctly diagnosed at least two of the three scenarios they were presented with and attributed the cause to mental illness. This is consistent with findings of previous studies that have reported college students are able to recognize symptoms of common mental health disorders such as depression, anxiety, and schizophrenia [47,48,49]. On the one hand, only 32% of undergraduate students in a study by Tai and Nguyen [50] were able to accurately diagnose depression for their vignette and instead identified the problem as “stress” which many respondents in our study indicated was the likely environmental causal factor for depression. Previous research suggests a complicated mix of genetic, biological, interpersonal, and environmental factors contribute to depression and other mental health disorders in college students [51,52,53].

The majority of respondents recommended professional help was needed by the students depicted in the vignettes; go to a counselor on campus, see a psychiatrist or psychologist, and talk with family and friends were the highest ranked choices. These findings contrast with other research reporting students prefer to seek religious counsel or talk to family or friend as opposed to seeking help through a mental health service or professional [7]. However, another study supports our findings. Reavley [28] found that general practitioners were selected as the most likely source of help by students If students are struggling with a mental health problem alone, studies suggest they are more likely to seek informal, non-professional, and online/self-help support rather than in-person professional help due to self or public stigma and social norms [12,54,55,56].

Seeking additional information about their mental health concerns online was supported by over half of our respondents. A review of online help-seeking activities by Pretorius, Chambers, and Coyle [57] reported text-based query via an internet engine was the most reported approach but social media, government or charity websites, live chat, instant messaging, and online communities were also used. The authors highlighted the advantages of anonymity and privacy, ease of access and immediacy, inclusivity, the ability to connect with others and share experiences, providing a gateway to further help-seeking, and a greater sense of control over their efforts. Online resources can play a role in early intervention if the information found online can help with early identification of concerning symptoms and assist in reaching out to mental health professional. Dunbar et al. [46] reported that students in their study were open to using online mental health treatment services (also known as telepsychology, e–mental health, and online therapy) versus in-person, even though utilization was low [58].

### 4.3. Risk of Mental Health Disorder Diagnosis

College students in our study were most at risk for anxiety, substance abuse, depression, sleep problems, and mania. Previous studies report high prevalence rates for anxiety, depressive and substance use disorders with associated quality of life impairment among college students [3,59,60]. In the context of the recent pandemic, Wang et al. [1] surveyed 2031 college students and found that 48% showed moderate-to-severe level of depression, 38% had a moderate-to-severe level of anxiety, and 18% reported suicidal thoughts in the two weeks preceding the survey. A significant majority indicated their stress/anxiety levels had increased during the pandemic. These findings highlight the mental health vulnerability of college students in general and the need for further study of interventions to enhance resilience in this population and provide access to information, support, and services for mental health concerns, especially in times of amplified stress.

### 4.4. Comparision of Mental Health Knowlege and Risk of Diagnosis by Levels of eMHL

When we examined the differences between levels of eMHL and knowledge, college students with high eMHL were more likely to accurately identify the mental health disorders and endorse professional help compared college students with low eMHL. High eMHL students were also less likely to be at risk of a mental health disorders. The implications of MHL are well documented. Studies show that college students with low MHL lack the skills to be able to identify mental health problems and seek professional help when necessary [12]. Further evidence suggests that lack of digital literacy skills in mental health have negative consequences on mental health, due to failure to identify safe web-based resources to support their self-management [36,37]. eMHL, while not extensively studied, may be viewed as a facet of MHL, and includes the skills to access and appraise evidence-based internet-based mental health information and resources to aid in the recognition, management, or prevention of mental health issues. Institutions with a mandate for mental health promotion among college students should take advantage of eMental health strategies that include interactive, customizable formats that provide easy access, greater variety, and rapid dissemination of information. Further research is needed on how eMHL impacts college students’ ability to identify relevant and reliable mental health information online and navigate the mental health care system.

### 4.5. Limitations

Limitations of the study were the small sample size and the use of mTurk to recruit participants and collect data. Although mTurk is used extensively in the social psychology research, there are innate limitations associated with this method of data collection, including participant inattention and compensation (self-selection bias). Another limitation was the use of vignettes representing symptomology of the various disorders to assess MHL; although the vignette method is helpful in providing a full description of symptoms, it is not an efficient and reliable method of evaluating a college students’ ability to recognize mental health issues, locate evidence-based resources, and take appropriate help-seeking action.

## 5. Conclusions

College students’ mental health issues are a growing concern on campuses across the United State due to short-term and extended consequences when mental health disorders are left untreated. MHL has emerged as an important concept because it is known to influence help-seeking for symptoms of mental health disorders, as well as attitudes toward mental health providers and compliance with treatment [48]. Many college students fail to seek help for psychiatric symptoms due to fear of personal stigma, not recognizing treatment is needed, and unawareness of available services. Findings of this study showed that eMHL scores were higher among women and those with a history of mental disorders. Most participants reported being diagnosed with at least two of the three mental disorders such as anxiety, depression, and schizophrenia. Additionally, college students in our study with high eMHL were more likely to accurately identify the mental health disorders and endorse professional help compared to college students with low eMHL. High eMHL students were also less likely to be at risk of a mental health disorders. The findings of this study add support to those of previous studies demonstrating that college students with high eMHL are more likely to recognize symptoms of common mental health disorders and recommend professional help compared to students with low eMHL. College students with higher eMHL are also at lower risk for a mental health disorder than low eMHL students. Our findings emphasize the need for targeted interventions to improve eMHL among college students with low eMHL to increase their understanding of symptoms and help-seeking options, as well as reduce stigma and enhance self-efficacy. Mental health interventions should be initiated early on during the university period to increase their understanding of how to achieve and maintain positive mental health, recognize mental health problems when they emerge, and actively seek treatment to minimize long-term consequences.

## Figures and Tables

**Table 1 healthcare-10-02406-t001:** Individual eHEALS items (N = 123).

Items		n	%
How useful do you feel the Internet is in helping you in making decisions about your mental health?	Not useful at all	0	0
	Not useful	4	3.3
	Unsure	27	22
	Useful	68	55.3
	Very Useful	24	19.5
How important is it for you to be able to access mental health resources on the Internet?	Not important at all	3	2.4
	Not important	10	8.1
	Unsure	15	12.2
	Important	51	41.5
	Very important	44	35.8
I know what mental health resources are available on the Internet	Strongly disagree	1	0.8
	Disagree	14	11.4
	Undecided	17	13.8
	Agree	65	52.8
	Strongly Agree	26	21.1
I know where to find helpful mental health resources on the Internet	Strongly disagree	0	0
	Disagree	16	13
	Undecided	12	9.8
	Agree	67	54.5
	Strongly Agree	28	22.8
I know how to find helpful mental health resources on the Internet	Strongly disagree	1	0.8
	Disagree	10	8.1
	Undecided	9	7.3
	Agree	69	56.1
	Strongly Agree	34	27.6
I know how to use the Internet to answer my questions about mental health	Strongly disagree	0	0
	Disagree	10	8.1
	Undecided	17	13.8
	Agree	66	53.7
	Strongly Agree	30	24.4
I know how to use the mental health information I find on the Internet to help decision	Strongly disagree	1	0.8
	Disagree	8	6.5
	Undecided	18	14.6
	Agree	65	52.8
	Strongly Agree	31	25.2
I have the skills I need to evaluate the mental health resources I find on the Internet	Strongly disagree	0	0
	Disagree	12	9.8
	Undecided	19	15.4
	Agree	60	48.8
	Strongly Agree	32	26
I can tell high quality mental health resources from low quality health resources on the Internet	Strongly disagree	4	3.3
	Disagree	5	4.1
	Undecided	25	20.3
	Agree	58	47.2
	Strongly Agree	31	25.2
I feel confident in using information from the Internet to make mental health decisions	Strongly disagree	3	2.4
	Disagree	15	12.2
	Undecided	23	18.7
	Agree	57	46.3
	Strongly Agree	25	20.3

**Table 2 healthcare-10-02406-t002:** Correctly diagnosed mental health conditions.

Scenario	Male		Female		Total
	Selected † n (%)	Correctly Diagnosed †† n (%)	Selected † n (%)	Correctly Diagnosed †† n (%)	Correctly Diagnosed †† n (%)
Depression	84 (68.3)	61 (73.5)	39 (31.7)	9 (23.1)	70 (48.8)
Anxiety	62 (50.4)	52 (83.9)	61 (49.6)	52 (85.2)	104 (84.6)
Schizophrenia	47 (38.2)	42 (89.4)	76 (61.8)	61 (80.3)	103 (83.7)

† Percentages are out of the total number of participants. †† Percentages are out of the total number who selected the scenario version.

**Table 3 healthcare-10-02406-t003:** Percentage of college students in each score category for each domain in the DSM-5 Self-Rated Level 1 Cross-Cutting Symptom Measure.

	None/ Not at All	Slight/Rare, Less than a Day or Two	Mild/ Several Days	Moderate/ More than Half the Days	Severe/ Nearly Every Day
	n (%)	n (%)	n (%)	n (%)	n (%)
Domain 1: Depression	42 (34.1)	29 (23.6)	23 (18.7)	21 (17.1)	8 (6.5)
Domain 2: Anger	43 (35)	36 (29.3)	27 (22)	13 (10.6)	4 (3.3)
Domain 3: Mania	51 (41.5)	23 (18.7)	26 (21.1)	14 (11.4)	9 (7.3)
Domain 4: Anxiety	29 (23.6)	24 (19.5)	32 (26)	21 (17.1)	17 (13.8)
Domain 5: Somatic Symptoms	51 (41.5)	30 (24.4)	13 (10.6)	17 (13.8)	12 (9.8)
Domain 6: Suicidal Ideation	91 (74)	10 (8.1)	13 (10.6)	5 (4.1)	4 (3.3)
Domain 7: Psychosis	97 (78.9)	8 (6.5)	11 (8.9)	4 (3.3)	3 (2.4)
Domain 8: Sleep Problems	55 (44.7)	19 (15.4)	22 (17.9)	18 (14.6)	9 (7.3)
Domain 9: Memory	67 (54.5)	29 (23.6)	13 (10.6)	6 (4.9)	8 (6.5)
Domain 10: Repetitive Thoughts or Behaviors	71 (57.7)	19 (15.4)	18 (14.6)	8 (6.5)	7 (5.7)
Domain 11: Dissociation	82 (66.7)	24 (19.5)	8 (6.5)	6 (4.9)	3 (2.4)
Domain 12: Personality Functioning	54 (43.9)	26 (21.1)	20 (16.3)	14 (11.4)	9 (7.3)
Domain 13: Substance Abuse	63 (51.2)	14 (11.4)	13 (10.6)	9 (7.3)	24 (19.5)

**Table 4 healthcare-10-02406-t004:** College students’ knowledge of mental health disorders by levels of eMHL.

Disorder		Depression		Anxiety		Schizophrenia	
eMental Health Literacy Group		Low n (%)	High n (%)	Low n (%)	High n (%)	Low n (%)	High n (%)
Correct Diagnosis		30 (48.4)	40 (65.6)	47 (75.8)	57 (93.4)	49 (79)	54 (88.5)
Primary Cause	Mental illness	8 (12.9)	13 (21.3)	29 (46.8)	38(62.3)	46 (74.2)	50 (82)
	Genetic factor	2 (3.2)	4 (6.6)	6 (9.7)	5 (8.2)	6 (9.7)	6 (9.8)
	Environmental factor	32 (51.6)	31 (50.8)	6 (9.7)	7 (11.5)	4 (6.5)	3 (4.9)
	Personal vulnerability	20 (32.3)	13 (21.3)	21 (33.9)	11 (18)	6 (9.7)	2 (3.3)
Actions to Take †	Talk with family and friends about what is bothering him/her	17 (27.4)	17 (27.9)	10 (16.1)	6 (9.8)	3 (4.8)	1 (1.6)
	See a psychiatrist/psychologist	19 (30.6)	16 (26.2)	39 (62.9)	47 (77)	55 (88.7)	55(90.2)
	Talk to a counselor on campus	22 (35.5)	28 (45.9)	10 (16.1)	7 (11.5)	0 (0)	5 (8.2)
Professional Help Needed	Yes	32 (51.6)	42 (68.9)	47 (75.8)	55 (90.2)	54 (87.1)	58 (95.1)
	No	15 (24.2)	9 (14.8)	8 (12.9)	5 (8.2)	3 (4.8)	2 (3.3)
	Undecided	15 (24.2)	10 (16.4)	7 (11.3)	1 (1.6)	5 (8.1)	1 (1.6)
Type of Professional Help	Primary care Provider	7 (21.9)	5 (11.9)	2 (4.3)	1 (1.8)	2 (3.7)	1 (1.7)
	Psychologist	13 (40.6)	23 (54.8)	23 (48.9)	23 (41.8)	17 (31.5)	13 (22.4)
	Psychiatrist	10 (31.3)	12 (28.6)	20 (42.6)	27 (49.1)	35 (64.8)	43 (74.1)
	Social Worker	1 (3.1)	1 (2.4)	2 (4.3)	3 (5.5)	0 (0)	1 (1.7)
	Other	1 (3.1)	1 (2.4)	0 (0)	1 (1.8)	0 (0)	0 (0)
Where to Find More Information	Self-help books	5 (8.1)	2 (3.3)	4 (6.5)	4 (6.6)	0 (0)	1 (1.6)
	Mental health websites	22 (35.5)	39 (63.9)	31 (50.0)	37 (60.7)	25 (40.3)	34(55.7)
	Campus library	3 (4.8)	1 (1.6)	4 (6.5)	1 (1.6)	5 (8.1)	1 (1.6)
	Someone with a similar problem	32 (51.6)	19 (31.1)	23 (37.1)	19 (31.1)	32 (51.6)	25 (41)

† One category is not presented because the category differed between scenarios. The percentages here will not sum to 100.

**Table 5 healthcare-10-02406-t005:** Risk of Diagnosis by low and high eMHL.

	Low Literacy	High Literacy	
Domain	n (%)	n (%)	*p **
Domain 1: Depression	28 (45.2)	24 (39.3)	0.514
Domain 2: Anger	23 (37.1)	21 (34.4)	0.757
Domain 3: Mania	26 (41.9)	23 (37.7)	0.632
Domain 4: Anxiety	38 (61.3)	32 (52.5)	0.323
Domain 5: Somatic Symptoms	22 (35.5)	20 (32.8)	0.752
Domain 6: Suicidal Ideation	21 (33.9)	11 (18)	0.045
Domain 7: Psychosis	15 (24.2)	11 (18)	0.403
Domain 8: Sleep Problems	26 (41.9)	23 (37.7)	0.632
Domain 9: Memory	15 (24.2)	12 (19.7)	0.545
Domain 10: Repetitive Thoughts or Behaviors	18 (29)	15 (24.6)	0.578
Domain 11: Dissociation	11 (17.7)	6 (9.8)	0.204
Domain 12: Personality Functioning	23 (37.1)	20 (32.8)	0.616
Domain 13: Substance Abuse	31 (50)	29 (47.5)	0.785

* *p*-values smaller than 0.004 are considered statistically significant.

## Data Availability

Not applicable.

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
