# Peer review of "College Students’ eMental Health Literacy and Risk of Diagnosis with Mental Health Disorders"

_healthcare, 2022, doi:10.3390/healthcare10122406_

Round 1

Reviewer 1 Report

The study addresses an interesting topic.

However, the study uses a sample that is too small for a quantitative study, so the results are not of interest to an international audience.

There are some methodological issues that should be reviewed.

Conclusions should be limited to the results of the study

Author Response

Dear reviewer 1 

I have attached the file. Please see the attached. Thanks

Reviewer 2 Report

a very extensive introduction including an analysis of the literature. Perhaps it would be worth adding a chapter of literature analysis and there to analyze the research activities carried out so far on this subject. The method part does not describe the research sample at all, says nothing about the students, why are they? what was the sample selection is it representative and for whom? for the country? There are no desirable research model or hypotheses. The authors write that the survey application provides for results, ie? in honor the results of the tebels are quite confusing. It is difficult to draw conclusions from this analysis. I recommend examining the coefficients of dependence, maybe c-pearson or kramer?

The discussion is again based on the analysis of the literature in this part is unacceptable.

Author Response

Dear reviewer 2

Thanks

Round 2

Reviewer 2 Report

Still not corrected:

The methodological part does not describe the research sample at all, does not say anything about the students, why?

they? How was the sample selected? Is it representative and for whom? for the country?

There is no desired research model or hypotheses.

Author Response

Dear reviewer,

Thanks for the comments.

Thanks
